

# Management of recurrent ureteral stricture: a retrospectively comparative study with robot-assisted laparoscopic surgery versus open approach

Qing Wang, Yuchao Lu, Henglong Hu, Jiaqiao Zhang, Baolong Qin, Jianning Zhu, Najib Isse Dirie, Zongbiao Zhang and Shaogang Wang

Department of Urology, Tongji Hospital, Tongji Medical College, Huazhong University of Science and Technology, Wuhan, Hubei, China

## ABSTRACT

**Background:** Management of recurrent ureteral stricture is challenging. Consensus on the best surgical choice has not been demonstrated. In this study, we aim to report our experience in treating recurrent ureteral stricture and demonstrate whether robot-assisted procedure for redo ureteral surgery is as effective as open procedure while remaining less invasive.

**Methods:** We retrospectively assessed 41 patients (22 robot-assisted surgeries and 19 open surgeries) who underwent consecutive robot-assisted and open procedures for redo ureteral surgery from January 2014 to 2018 in our institution. Perioperative outcomes, including demographics, operative time, estimated blood loss, complications, pain scores, success rate and cost, were compared between two groups.

**Results:** There was no significant intergroup difference in terms of age, body mass index, gender composition and American Society of Anesthesiologists scores. A total of 31 patients underwent redo pyeloplasty and ten underwent redo uretero-ureterostomy. Compared with open group, robot-assisted group showed shorter operative time (124.55 min vs. 185.11 min, $p < 0.0001$), less estimated blood loss (100.00 mL vs. 182.60 mL, $p = 0.008$) and higher cost (61161.77¥ vs. 39470.79¥, $p < 0.0001$). Complication rate and pain scores were similar between two groups. Median follow-up periods were 30 and 48 months for robot-assisted and open group respectively. Success rate in the robot-assisted (85.71%) and the open group (82.35%) was not significantly different.

**Conclusions:** Robot-assisted surgery for recurrent stricture after previous ureteral reconstruction is as effective as open procedure and is associated with shorter operative time and less estimated blood loss.

Corresponding authors
Zongbiao Zhang, zzb070@126.com
Shaogang Wang,
sgwangtjm@163.com

## INTRODUCTION

Management of recurrent stricture after previous ureteral reconstruction is challenging, both in terms of decision-making and surgical technique. Endopyelotomy is the most minimally invasive endourological choice for recurrent ureteropelvic junction obstruction

(UPJO). However, the reported success rate (39–83.5%) seems to be suboptimal (*Braga et al., 2007*; *Patel et al., 2011*; *Abdrabuh et al., 2018*). Hence, there comes again an interest in secondary reconstruction. Redo ureteral surgery is difficult due to scar formation, altered anatomic planes and decreased vascularity of the ureter. In such cases, open surgery is an excellent choice and the success rate of open redo pyeloplasty reaches up to 80–100% (*Abdel-Karim et al., 2016*; *Piaggio, Noh & González, 2007*; *Vannahme et al., 2014*). Open procedure has been then suggested as the gold standard for recurrent UPJO. Redo laparoscopic pyeloplasty also shows comparable outcomes to open procedure (*Moscardi et al., 2017*; *Powell et al., 2015*). However, laparoscopic procedure is more technically difficult and requires a longer learning curve, which limits its widespread use in redo ureteral surgery.

The da Vinci robot assisted system, which shows advantages of three-dimensional vision, tremor filtering and seven degrees of freedom, has been increasingly used in ureteral reconstruction (*Minnillo et al., 2011*; *Di Gregorio et al., 2014*). In a large series conducted by *Buffi et al. (2017)* they find that robotic surgery for benign ureteral stricture and UPJO is associated with low risk of high-grade complications and good outcomes. Subsequently, more and more studies have demonstrated that robot-assisted redo pyeloplasty is minimally invasive and effective for recurrent UPJO, with excellent success rate ranging from 88% to 100% (*Lindgren et al., 2012*; *Asensio et al., 2015*; *Davis et al., 2016*).

To the best of our knowledge, there has been no study comparing the outcomes of robot-assisted procedure with open procedure in redo ureteral surgery. Moreover, most previous studies focus on pediatric failed pyeloplasty, while adult cases and cases involving recurrent stricture in other sites of the ureter are rarely reported. In the present study, outcomes of robotic and open procedure in redo pyeloplasty and uretero-ureterostomy were compared. We aim to report our experience in treating recurrent ureteral stricture and provide urologists with some evidence for surgical decision.

## MATERIALS AND METHODS

We retrospectively identified consecutive robot-assisted and open procedures performed from January 2014 to 2018 in our institution for recurrent stricture with previous ureteral reconstruction. Patients treated with a "simple" redo ureteral anastomosis were included. Preoperative clinical assessments included evaluation of symptoms and computed tomography (CT), urography, magnetic resonance urography or intravenous urography. Indications for surgery included persistent clinical symptoms or worsening hydronephrosis on imaging or with worsening renal function. All robot-assisted procedures were performed by one experienced urologist (WSG) using the da Vinci Surgical System (Intuitive Surgical, Sunnyvale, CA, USA). Open surgeries were performed by two experienced surgeons (including WSG). The choice between a robot-assisted and an open approach was based on the surgeon's preference and the patient's choice. Follow-up assessments were conducted in the third and sixth months after removing double J stent and then once every year. Success was defined as a stable or reduced degree of hydronephrosis on ultrasound or CT with symptom relief. Complications were

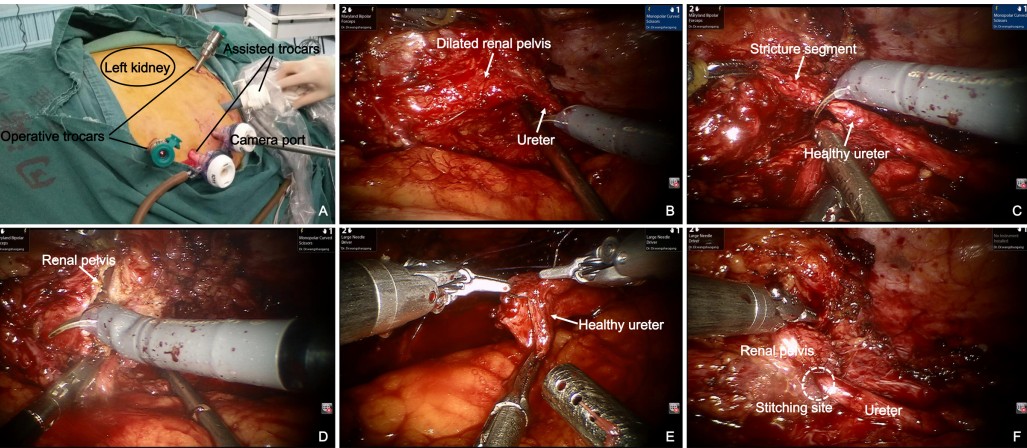

**Figure 1 Robot-assisted laparoscopic redo pyeloplasty for a 59-year-old male who underwent a failed left pyeloplasty before.** (A) Modified 45° lateral decubitus position for upper ureter stricture repair. Location of trocars is shown in this image. (B) The stricture segment was identified according to intraoperative morphological changes (dilated renal pelvis to normal ureter). (C) Excision of the stricture segment. (D) Removal of redundant renal pelvis and fibrosis segment. (E) Anastomosis was performed with fine interrupted suture. (F) Completed end-to-end anastomosis.

classified using the Clavien–Dindo grading system (*Dindo, Demartines & Clavien, 2004*). Perioperative outcomes, including operative time, estimated blood loss, complications, pain scores, cost and success rate were compared between two groups. We did not include the postoperative stay to analyze it as a variable, because it was not valuable due to the hospital discharge policy of our institution.

This study has been approved by the Institutional Review Board of Tongji Hospital, Tongji Medical College, Huazhong University of Science and Technology (2019S935). Absence of informed consent was approved because of the retrospective nature of the study. In addition, the data were analyzed anonymously.

## Procedures

Patients underwent dismembered pyeloplasty for UPJO and uretero-ureterostomy for proximal ureter strictures (ureteral pelvic junction to the external iliac vessels). All robotic procedures were performed transperitoneally. Patients were placed in a modified 45-degree lateral decubitus position after general anesthesia. A periumbilical 12 mm camera port, two 8 mm robotic trocars oriented toward the stricture and two 12 mm assistant ports were placed for the operation (Fig. 1). Iliac vessels, gonadal veins and paracolic sulcus were all important anatomical markers to quickly identify ureter or renal pelvis. The stricture segment was identified according to preoperative imaging localization and intraoperative morphological changes (dilated to normal ureter). After excising the stricture segment, proximal ureter was incised along the longitudinal axis of ureter to renal pelvis or at least 1 cm normal ureter was visualized. The normal distal ureter was also incised for 1 cm. Once both ends of the ureter were adequately trimmed to healthy areas and correctly oriented, end-to-end reanastomosis was started laterally and performed in an interrupted fashion with a 4-0 polyglycolic acid suture. A double J ureteral stent was

normally placed in our institution for about 4 weeks and a drain tube was also placed. The Foley catheter was placed for 2–3 days after surgery and the drainage tube was removed on the following day if there was no increase in drain output.

Surgical details of open procedure have been previously described by other investigators (*Nakada & Sara, 2016*). General principles were similar to those described for the robot-assisted procedure.

### Statistics

Statistical analysis was performed with SPSS for Windows version 19.0 (IBM Corporation, Armonk, NY, USA). For variables with normal distribution, data were presented as mean ± standard deviation (SD) and compared using Student's *t*-test. For variables with a non-normal distribution, data were presented as median (range) and were compared using the Mann–Whitney *U*-test. Categorical variables were compared by Chi-squared test or Fisher's exact test.

## RESULTS

A total of 41 patients underwent redo ureteral reconstruction in our institution, including 22 robot-assisted and 19 open cases. Most patients came to hospital with flank pain while some recurrence was asymptomatic. Three patients presented with fever due to urinary infection. Hematuria was detected in one patient and the stricture was confirmed to be benign with pathological assessment. Initial etiologies for ureteral stricture were clarified into congenital, stone disease, vascular anomalies and polyp. Median interval between initial and redo surgery was 8 years (range, 1–25 years) in the robot-assisted group and also 8 years (range, 0.5–31 years) in the open group. Primary surgery for ureteral stricture consisted of 27 open procedures and 14 laparoscopic procedures. Missed lower pole crossing vessels in initial surgery were found in two patients. Some patients had once chosen endourological techniques after failed initial surgery, such as double J stent, balloon dilation and endopyelotomy. Most (80.9%) patients directly underwent redo surgery. Moreover, one patient in the open group had undergone two open reconstructions previously (Table 1).

No significant difference was found in terms of age, body mass index (BMI), gender composition, disease side and American Society of Anesthesiologists (ASA) scores between two groups. In our series, 19 patients underwent redo pyeloplasty for recurrent UPJO and three patients underwent redo uretero-ureterostomy for recurrent proximal ureteral stricture with robotic procedure. None of the robot-assisted operations necessitated an open conversion. Correspondingly, 12 patients underwent redo pyeloplasty and seven patients underwent redo uretero-ureterostomy in the open group. There were no intraoperative complications in both groups. We encountered many challenges during dissection of periureteral planes due to significant fibrosis and adhesions in most patients. Compared with open group, robotic group showed shorter operative time (124.55 min vs. 185.11 min, *p* < 0.0001) and less estimated blood loss (100.00 mL vs. 182.6 mL, *p* = 0.008). Postoperative pain was assessed on the first day using visual analogue scores (VAS) and no significant difference was observed in the pain scores between two groups.

**Table 1 Description of symptoms, recurrent interval, etiology for initial stricture, and primary procedures for patients.**

| Item | Robot ($n$ = 22) | Open ($n$ = 19) |
|---|---|---|
| **Symptoms** | **$n$ (%)** | |
| Flank pain | 14 (63.64) | 9 (47.37) |
| Fever | 1 (4.54) | 2 (10.53) |
| Haematuria | 0 | 1 (5.26) |
| Asymptomatic | 7 (31.82) | 7 (36.84) |
| **Recurrent interval (years)** | **Median (range)** | |
| | 8 (1–25) | 8 (0.5–31) |
| **Etiologies for initial UPJO** | **$n$ (%)** | |
| Congenital | 13 | 8 |
| Stone disease | 6 | 2 |
| Vascular anomaly | 0 | 2 |
| **Etiologies for initial ureteral stricture** | **$n$ (%)** | |
| Stone disease | 3 | 6 |
| Ureteral polyp | 0 | 1 |
| **Primary procedures** | **$n$ (%)** | |
| Open pyeloplasty | 9 | 6 |
| Open uretero-ureterostomy | 1 | 5 |
| Laparoscopic pyeloplasty | 5 | 4 |
| Laparoscopic uretero-ureterostomy | 1 | 2 |
| Open pyeloplasty + nephrostomy | 1 | 1 |
| Open uretero-ureterostomy + balloon dilation | 1 | 0 |
| Open pyeloplasty + endopyelotomy | 1 | 0 |
| Open pyeloplasty + double J stent | 2 | 0 |
| Laparoscopic pyeloplasty + nephrostomy | 1 | 0 |
| Laparoscopic pyeloplasty + balloon dilation + endopyelotomy | 0 | 1 |

Postoperative complication rate was similar between two groups. Although more patients suffered from fever (Grade II) in the open group, no significant difference was detected (36.84% vs. 9.09%, $p$ = 0.057). One patient suffered leakage of urine (Grade IIIa) and one showed incision hematoma (Grade IIIb) after open surgery. The patient with urine leakage experienced longer drainage time and longer double J insertion. The other patient underwent surgical intervention for the incision hematomaobotic. In addition, robot-assisted group required higher cost (61161.77¥ vs. 39470.79¥, $p$ < 0.0001) (Table 2).

## Follow up

One patient in robot-assisted group and two in open group were lost to follow-up. The median follow-up duration was 30 months (range, 19–48 months) for the robot-assisted group and 48 months (range, 20–63 months) for the open group. There is significant difference in terms of follow-up length between two groups ($p$ = 0.001). Two patients in the open group showed more severe hydronephrosis in the sixth and 12th months after surgery. Both of them were treated with double J stent placement but

**Table 2  Preoperative characteristics and postoperative outcomes of patients.**

| Item | Robot (n = 22) | Open (n = 19) | p value |
|---|---|---|---|
| **Mean age (years)** | **Mean ± SD (median)** | | 0.760 |
| | 37.82 ± 18.56 (36) | 36.16 ± 15.54 (39) | |
| **Mean BMI (kg/m²)** | **Mean ± SD (median)** | | 0.969 |
| | 22.75 ± 3.65 (22.48) | 22.79 ± 2.94 (22.86) | |
| **Gender** | **n (%)** | | 0.737 |
| Male | 16 (72.73) | 12 (63.14) | |
| Female | 6 (27.27) | 7 (36.86) | |
| **Side of disease** | **n (%)** | | >0.999 |
| Left | 14 (63.64) | 12 (63.16) | |
| Right | 8 (36.36) | 7 (36.84) | |
| **ASA** | **n (%)** | | 0.742 |
| I | 5 (22.73) | 4 (21.05) | |
| II | 15 (68.18) | 14 (73.68) | |
| III | 2 (9.09) | 1 (5.27) | |
| **Procedure** | **n (%)** | | 0.145 |
| Pyeloplasty | 19 (86.36) | 12 (63.16) | |
| Uretero-ureterostomy | 3 (13.63) | 7 (36.84) | |
| **Mean operative time (min)** | **Mean ± SD** | | <0.0001 |
| | 124.55 ± 48.45 | 185.11 ± 49.71 | |
| **Mean blood loss (ml)** | **Mean ± SD** | | 0.008 |
| | 100.00 ± 18.43 | 182.60 ± 23.89 | |
| **Postoperative complications*** | **n (%)** | | |
| Grade II | 2 (9.09) | 7 (36.84) | 0.057 |
| Grade IIIa | 0 | 1 (5.26) | 0.463 |
| Grade IIIb | 0 | 1 (5.26) | 0.463 |
| **Median VAS** | **Median (range)** | | 0.053 |
| | 2.5 (1–5) | 2 (1–3) | |
| **Cost (¥)** | **Mean ± SD** | | <0.0001 |
| | 61161.77 ± 8567.67 | 38470.79 ± 9764.00 | |

Note:
BMI, Body mass index; ASA, American Society of Anesthesiologists Score; VAS, Visual analogue score; SD, Standard deviation.
* Postoperative complications were classified using the Clavien–Dindo grading system.

outcomes were not ideal. One patient underwent nephrectomy and the other underwent dialysis. For patients with stable hydronephrosis, we assessed their renal function via routine biochemical analysis and all of them showed stable renal function. The success rate in the robot-assisted and open groups was 85.71% and 82.35% respectively, with no statistical difference between two groups (Table 3).

## DISCUSSION

It is reported that approximately 11% of the children who underwent pyeloplasty require a secondary procedure in the United States (Dy et al., 2016). Signs of recurrent obstruction

**Table 3 Follow-up outcomes of patients.**

| Items | Robot (*n* = 21) | Open (*n* = 17) | *p* value |
|---|---|---|---|
| **Median follow up (month)** | **Median (range)** | | 0.001 |
| | 30 (19–48) | 48 (20–63) | |
| **Success**[*] | ***n* (%)** | | 0.775 |
| | 18 (85.71%) | 14 (82.35%) | |
| **Outcomes** | ***n* (%)** | | |
| Decreased hydronephrosis | 14 (66.67) | 12 (70.59) | – |
| Stable hydronephrosis + symptom resolution | 4 (19.05) | 2 (11.76) | – |
| Stable hydronephrosis + unrelieved symptoms | 3 (14.28) | 1 (5.89) | – |
| Increased hydronephrosis | 0 | 2 (11.76) | – |

Note:
[*] Success was defined as a stable or decreased degree of hydronephrosis and absence of symptoms.

are variable. *Jacobson et al. (2019)* report that 36% patients with recurrent UPJO showed symptoms of pain, urinary infection or hematuria in their series. In our study, 56.09% patients came to the hospital with flank pain while 34.15% without symptoms and just showed a progressive hydronephrosis. Recurrence appears early after the first intervention in many studies. However, recurrence might occur until several years after primary repair in our series, indicating a need for long-term follow-up of these patients. We found that the most common cause for recurrent stricture was periureteric fibrosis or scars caused by urinary extravasation, which restricted peristalsis of the ureter. Missed lower pole crossing vessels was also another reason for failed pyeloplasty. In an anatomic analysis of 146 endocasts of kidney collecting system together with intrarenal arteries and veins, *Sampaio & Favorito (1993)* found that 65.1% of the cases showed renal vessels on the anterior surface of the ureteropelvic junction. When pyeloplasty is done via retroperitoneal approach, there is a limited view of the anterior surface and the anterior crossing vessels might be missed.

Studies have shown that open redo pyeloplasty is associated with excellent outcomes for recurrent stricture. Redo laparoscopic pyeloplasty is also an effective treatment for recurrent UPJO, with reported success rate of 80–100% (*Moscardi et al., 2017*; *Powell et al., 2015*). *Abdel-Karim et al. (2016)* demonstrate similar success rate between open and laparoscopic redo pyeloplasty. Although laparoscopic technique is associated with shorter hospital stays and less postoperative pain, it is more difficult and involves longer operative time and more blood loss. Similarly, *Sundaram et al. (2003)* and *Nakada, McDougall & Clayman (1995)* both report in their early series that redo laparoscopic pyeloplasty requires a long operative time (average 6.2 and 9 h respectively).

Introduction of robotic systems has overcome limitations of laparoscopy. Robot-assisted redo pyeloplasty has been reported to be an excellent choice for recurrent UPJO. *Niver et al. (2012)* report that robotic reoperation is as safe and effective as primary operation for UPJO. *Atug et al. (2006)* note a success rate of 100% in one of the first pediatric series of robot-assisted redo pyeloplasty. *Lindgren et al. (2012)* conduct robot-assisted redo pyeloplasty in 16 pediatric patients and 88% patients show

postoperative improvement on radiographic findings. *Hemal et al. (2008)* (9 patients) also report a success rate of 100% for redo robot-assisted pyeloplasty. Correspondingly, we achieved a success rate of 85.71% for robot-assisted procedure in the present study. Three patients showed stable hydronephrosis and unrelieved symptoms after robot-assisted surgery, but their renal function remained stable during the follow-up period. No patient in the robot-assisted group required further intervention.

To our knowledge, this study represented the first attempt to compare the outcomes of robot-assisted and open procedure in treating recurrent benign ureteral stenosis of the UPJ and the ureter. We found that robotic assistance significantly reduced the operative time and estimated blood loss for redo ureteral surgeries, which showed advantages of minimal invasion. Benefits of robotic assistance were seen in easier meticulous dissection, better delineation of previous scarred tissue and preservation of the periureteral sheath containing blood supply to ureter. *Lee et al. (2006)* demonstrate shorter hospital stay and similar success rate for robot-assisted procedure compared to open procedure in treating primary UPJO. They also indicate that although robot-assisted procedure requires longer operative time (219 min vs. 181 min), it improves and approaches the operative time of open procedure as the operator's experience increases. *Isac et al. (2013)* and *Kozinn et al. (2012)* also reported similar findings in their experience with robotic and open ureteroneocystostomy.

No significant difference in terms of postoperative complication rate between robotic and open group was observed in our study, which is similar with what *Abdel-Karim et al. (2016)* find in their series of laparoscopic vs. open redo pyeloplasty. This may be partly due to the small sample size in our study. However, it should be indicated that more patients suffered fever and urine leakage and incision hematoma were observed in the open group, indicating a more invasive character of the open procedure. One feature of robot-assisted surgery observed in our study was that this technique was more expensive. Patients in robot-assisted group spent an average of 20,000¥ more than those in open group and this may influence the choice of surgical approach among patients.

There are several limitations in our study. Firstly, it is a relatively small retrospective case series with selection bias. A larger number of patients should be involved in the future to verify current findings. Secondly, we don't conduct renal emission CT to assess the affected side renal function in all patients. Thirdly, there is significant difference in terms of follow-up length between robotic and open group, a longer follow-up time is required to confirm the success rate.

## CONCLUSIONS

Robot-assisted procedure for recurrent stricture after previous ureteral reconstruction is as effective as open procedure. Moreover, robot-assisted technique shows decreased blood loss and less operative time, which provides advantages of minimally invasive surgery.

## ABBREVIATIONS

**UPJO**      Ureteropelvic junction obstruction
**CT**      Computed tomography

| BMI | Body mass index |
| VAS | Visual analogue scores |
| ASA | American society of anesthesiologists. |

### Funding
This work was supported by the National Key Research and Development Program of China (No. 2016YFC0902601). The funders had no role in study design, data collection and analysis, decision to publish, or preparation of the manuscript.

### Grant Disclosures
The following grant information was disclosed by the authors:
National Key Research and Development Program of China: 2016YFC0902601.

### Competing Interests
The authors declare that they have no competing interests.

### Author Contributions
- Qing Wang performed the experiments, analyzed the data, contributed reagents/materials/analysis tools, authored or reviewed drafts of the paper, approved the final draft.
- Yuchao Lu analyzed the data, prepared figures and/or tables, approved the final draft.
- Henglong Hu analyzed the data, contributed reagents/materials/analysis tools, prepared figures and/or tables, approved the final draft.
- Jiaqiao Zhang analyzed the data, contributed reagents/materials/analysis tools, prepared figures and/or tables, approved the final draft.
- Baolong Qin performed the experiments, prepared figures and/or tables, approved the final draft.
- Jianning Zhu analyzed the data, prepared figures and/or tables, approved the final draft.
- Najib Isse Dirie conceived and designed the experiments, authored or reviewed drafts of the paper, approved the final draft.
- Zongbiao Zhang conceived and designed the experiments, authored or reviewed drafts of the paper, approved the final draft.
- Shaogang Wang conceived and designed the experiments, authored or reviewed drafts of the paper, approved the final draft.

### Human Ethics
The following information was supplied relating to ethical approvals (i.e., approving body and any reference numbers):

This study has been approved by the Institutional Review Board of Tongji Hospital, Tongji Medical College, Huazhong University of Science and Technology (2019S935).
## Data Availability

All data generated or analyzed are available in the Supplemental File.

## Supplemental Information

Supplemental information for this article can be found online at http://dx.doi.org/10.7717/peerj.8166#supplemental-information.

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
