# Peer review of "Management of recurrent ureteral stricture: a retrospectively comparative study with robot-assisted laparoscopic surgery versus open approach"

_PeerJ, doi:10.7717/peerj.8166_

## Round 0.1 · original submission · Major Revisions

Please respond to the reviewers comments!

thanks

best regards

Reviewer 1 ·

Basic reporting

The standard reporting is fine.

Experimental design

-

Validity of the findings

See my general comments.

Additional comments

Authors performed a retrospective study upon their experience in treating recurrent benign ureteral stricture. The tested the hypothesis that redo ureteral reconstruction is as effective as open procedure. They considered 41 patients comparing 22 robot-assisted surgeries with 19 open surgeries. The study was not a RCT. They found no significant intergroup difference in terms of age, gender composition and ASA scores. Robot-assisted group showed shorter operative time, less estimated blood loss, shorter hospital stays and higher cost. Similar success rates: in the robot-assisted (85.71%) and the open group (82.35%). Authors concluded that robot-assisted procedure for recurrent stricture after previous ureteral reconstruction is as effective as open procedure and is associated with shorter operative time, less estimated blood loss and
shorter hospital stays, but higher cost.
This is an interesting study, but there are some flaws: the sample is not homogeneous (Initial etiologies for ureteral stricture were clarified into congenital, stone disease, vascular anomalies and polyp), the study is retrospective and not RCT and no clear indication to open vs. robotic was cleared in the manuscript. We would like to see in the follow-up the assessment of renal function. Is it known preoperatively?

Reviewer 2 ·

Basic reporting

Clear and unambiguous, professional English used throughout.
The article must be written in English and must use clear, unambiguous, technically correct text. The article must conform to professional standards of courtesy and expression.

Reviewer considerations:
In the PDF document of the authors themselves, there are notes throughout the document where, in my opinion, the writing needs to be improved to make it more understandable.

Title of the manuscript is redundant in the text, it can be improved:
Current:
Management of recurrent stricture after previous ureteral reconstruction: Robot-assisted laparoscopic surgery versus open surgery

Suggestion:
Management of recurrent ureteral stricture: "Comparative study with robot-assisted laparoscopic surgery versus open approach"

Abstract:
Background:
To report our experience in treating recurrent benign ureteral stricture and demonstrate whether robot-assisted procedure for redo ureteral reconstruction is as effective as open procedure while remaining less invasive.

They wrote it down here it is only the purpose of the study and prior to this lack a short but important text as background.

Methods and Results: Both sections of the Abstract should reflect the type of patients the study is about: most recurrent UP obstruction and less recurrent ureteral stenosis.

Conclusions: According to the instructions for the authors, Conclusions are not included in the ABSTRACT, only until after the Discussion.

Subjets: Lack.
Keywords: Lack.

The english language check of:
Introduction to References: In the same PDF of the authors' manuscript.

Literature references, sufficient field background/context provided.
The article should include sufficient introduction and background to demonstrate how the work fits into the broader field of knowledge. Relevant prior literature should be appropriately referenced.

Reviewer considerations:
In the introduction I suggest including relevant information from more publications in the adult population, since most of their references in children and mainly about pyeloplasty.

Professional article structure, figures, tables. Raw data shared.

Reviewer considerations:
The instructions for authors according to the suggested format, some aspects are missing, that in the authors' own PDF register notes clarifying and sometimes deleting sections.

Figures should be relevant to the content of the article, of sufficient resolution, and appropriately described and labeled.

Reviewer considerations:
Figure 1. Improve the description of their images:
(A) Modified 45 ° lateral decubitus position for upper ureter stricture repair.This image shows above all the location of the trocars.

(B-D) Dissection and transection of the ureter. There are 3 transoperative images, where they should highlight the characteristics of the surgical anatomy and details of the surgical technique they used.
Further:
Images D, E and F are not good images, since they do not adequately show the transoperative details of anatomy and surgical technique. I suggest changing them for other better images.

Table 1. The title is incomplete, it must describe the information contained in the table.
I suggest to better organize the presentation of the data in the table and if necessary use table footers in "Items" that are extensive.
For example: Symptoms, n (%), n (%) should be placed on the column of cases and percentage, also median (range), and "n".
It is also convenient to adjust the percentages to 100% in both columns of Symptoms.
Item "History", I suggest changing it to: Prior or Primary Procedures.

Tables 2 and 3: Also their titles are incomplete and it is also necessary to better organize the presentation of the data.
Table 2. In Item “Postoperative complications”, at the table footer, inform that the classification of the degrees of complications are based on the Clavien and Dindo Classification.
Further:
Table 2. Indicate that the Items: age, operative time, blood loss and hospital stays, in their results include the standard deviation


All appropriate raw data have been made available in accordance with our Data Sharing policy.

Reviewer considerations:

The raw data (or code) be opened, and is well described in English., Only the abbreviation VAS, which is located in column "P" of the spreadsheet does not guess the meaning.


Self-contained with relevant results to hypotheses.

Reviewer considerations:
The variables analyzed and the results are adequate to respond to the hypothesis raised, and in the same PDF of the authors there are notes where its writing needs to be improved.

Experimental design

Original primary research within Aims and Scope of the journal.
Research question well defined, relevant & meaningful. It is stated how research fills an identified knowledge gap.

Reviewer considerations:

This is the aim:
In the current study, we aim to report our experience in treating recurrent UPJO or ureteral strictures and demonstrate whether robot-assisted repair in redo ureteral reconstruction is as safe and effective as open repair.
I consider that the research question is correct.

Rigorous investigation performed to a high technical & ethical standard.

Reviewer considerations:
This study is retrospective, observational and comparative, it was approved by the ethics committee of hospital institution. Therefore, it does not violate ethical principles.

Methods described with sufficient detail & information to replicate.

Reviewer considerations:

The methods are described enough to reproduce the study.

Validity of the findings

Impact and novelty not assessed. Negative/inconclusive results accepted. Meaningful replication encouraged where rationale & benefit to literature is clearly stated.

Reviewer considerations:
The results obtained by the researchers support the hypothesis raised in the objective of the investigation, I only consider that the days of postoperative stay, for the reason stated by the researchers themselves, depending on the institutional care policy, I suggest they should eliminate it, to avoid invalid conclusions.

All underlying data have been provided; they are robust, statistically sound, & controlled.

Reviewer considerations:
The limitations of the study are recognized by the researchers, the most important being: limited number of cases of both study groups and the significant difference in follow-up time between both study groups.

Conclusions are well stated, linked to original research question & limited to supporting results.

Reviewer considerations:

The conclusions are appropriate and consistent with the research question and are supported by the results, except for:the statement of:shorter hospital stays.That it contrasts with the results obtained in the investigation and that in the discussion they comment on it, so, I suggest you should not take this data into account (consider it not valuable).

Additional comments

As a general recommendation, researchers should avoid issuing personal opinions.

In the PDF document of the same researchers, notes with corrections and suggestions to improve the manuscript are indicated.

Attachment of the Reviewer:
The PDF of the researchers themselves.

https://www.ncbi.nlm.nih.gov/pubmed/?term=recurrent+UPJO+%2B+Robotic+surgery/ recurrent UPJO + Robotic surgery
Publications 20 (adults and children).

https://www.ncbi.nlm.nih.gov/pubmed/?term=recurrent+UPJO+%2B+Robotic+surgery/ recurrent ureteral obstruction + robotic surgery
Publications 30 (adults and children)… Several duplicate publications.

https://www.ncbi.nlm.nih.gov/pubmed/?term=recurrent+UPJO+%2B+Robotic+surgery/recurrent ureteral obstruction + open surgery
Publications 112 (adults and children). Several of these useful and interesting.


Annex:
Abstract of a comparative study of robot versus open surgery in children.

J Pediatr Urol. 2015 Apr;11(2):69.e1-6. doi: 10.1016/j.jpurol.2014.10.009. Epub 2015 Feb 24.
Failed pyeloplasty in children: Is robot-assisted laparoscopic reoperative repair feasible?
Asensio M1, Gander R2, Royo GF1, Lloret J3.

Abstract
OBJECTIVE:
In this study we aim to demonstrate that robot-assisted laparoscopic (RAL) reoperative repair is safe and effective and even less technically demanding than open repair for recurrent ureteropelvic-junction obstruction (UPJO).
STUDY DESIGN:
A retrospective study was conducted of all cases of failed open pyeloplasties who underwent RAL reoperative repair at our institution between January 2010 and December 2013. The general surgical procedure was the same we previously described for robot-assisted laparoscopic pyeloplasty. Success was defined as: improvement in the degree of hydronephrosis at ultrasound, improvement of diuretic washout time at postoperative diuretic renogram (<15 min), improvement or at least stable differential renal function and absence of symptoms. These radiographic and symptomatic criteria of success were considered the primary outcomes. Secondary outcomes included complications and length of hospital stay.
RESULTS:
Between 2000 and 2013 a total of 153 patients underwent open Anderson-Hynes dismembered pyeloplasty. Of these 9 (6%) had recurrent UPJO. Four patients underwent open redo pyeloplasty. As a result, our study population comprised 5 children who underwent reoperative RALP repair. Patient characteristics and outcomes are summarized in the table below. Our success rate was 100%.
DISCUSSION:
Due to the low failure rate of open dismembered pyeloplasty there is no consensus on the best surgical approach for recurrent obstruction. While endoscopic approaches have been favored in adults, children have shown better success rates with repeat pyeloplasty. Laparoscopic salvage pyeloplasty for failed open procedures has become more popular and has been shown to result in excellent outcomes while providing the advantages of minimally invasive surgery. To date, the literature regarding the use of RALP for failed open procedures in the pediatric population is scarce. Only 2 pediatric series of robotic reoperative pyeloplasty have been reported by Helmal et al. (9 patients) and Lindgren et al. (16 patients) with a success rate of 100 and 88%, respectively. Although this is one of the first published studies about robot-assisted laparoscopic reoperative repair for failed open pyeloplasty in pediatric patients, we acknowledge the limitations of our study due to the small number of patients, its retrospective nature and limited follow-up time.
CONCLUSIONS:
The incidence of failed open pyeloplasty is as low as 5% and management remains controversial. As reported by other authors, we believe that crossing vessels play a particularly important role in secondary obstruction and adversely impact the outcome. Redo pyeloplasty, open or minimally invasive, is associated with high success rates (80-100%) and therefore considered the treatment of choice by the majority of authors nowadays. Additionally, RALP for secondary procedures has demonstrated to be safe and even less technically demanding when compared to the open approach, providing the advantages of minimally invasive surgery.


Others References:

Springerplus. 2014 Oct 3;3:580. doi: 10.1186/2193-1801-3-580. eCollection 2014.
Passing from open to robotic surgery for dismembered pyeloplasty: a single centre experience.
Di Gregorio M1, Botnaru A1, Bairy L2, Lorge F1.

FINDINGS:
From 17 patients who underwent Da Vinci Robot procedure, 15 followed the complete treatment (2 were converted to laparotomy). Two patients had post-operative urine leakage; the stent was changed under sedation without further sequelae. The mean operative time was 189 minutes. The average hospital stay was 4 days. The average follow-up was 25 months. There was only one patient with UPJ stenosis at 6 months and he was treated by balloon dilation. All patients were followed with MAG 3 lasix renal scan, CT or urography. Except the patient with recurrent stenosis, all patients were asymptomatic without objective evidence of obstruction at the present time.
CONCLUSIONS:
Robotic pyeloplasty technique is feasible and gives good results without previous laparoscopic experience.







J Urol. 2011 Apr;185(4):1455-60. doi: 10.1016/j.juro.2010.11.056. Epub 2011 Feb 19.
Long-term experience and outcomes of robotic assisted laparoscopic pyeloplasty in children and young adults.
Minnillo BJ1, Cruz JA, Sayao RH, Passerotti CC, Houck CS, Meier PM, Borer JG, Diamond DA, Retik AB, Nguyen HT.

Abstract
PURPOSE:
Laparoscopic pyeloplasty is one of the more common robotic assisted procedures performed in children. However, data regarding long-term experience and clinical outcomes for this procedure are limited. We evaluated the long-term outcomes in a large series of patients undergoing robotic assisted laparoscopic pyeloplasty at a teaching institution, and the effect of a collaborative program between the robotic surgeons, surgical nurses and anesthesiologists on overall operative time.
MATERIALS AND METHODS:
We retrospectively reviewed 155 patients who underwent robotic assisted laparoscopic pyeloplasty between 2002 and 2009. Operative data, including surgical approach, type of procedure, total and specific operative times and placement of ureteral stents, were determined. Postoperative outcome measurements, including duration of hospital stay, duration of Foley catheter drainage, radiological findings and any subsequent complications, were assessed.
RESULTS:
Mean operative time and length of hospitalization decreased significantly by the end of the study. At a mean followup of 31.7 months the primary success rate was 96% (hydronephrosis was improved in 85% of patients and stable in 11%). The complication rate was 11%, and recurrent obstruction requiring redo robotic assisted laparoscopic pyeloplasty or open pyeloplasty developed in 3% of patients. Success rate and complication rate were unchanged during the study period.
CONCLUSIONS:
This study confirms that even within the confines of a pediatric urology training program successful collaboration between robotic surgeons, surgical nurses and anesthesiologists can lead to shorter operative times and hospital stays. Long-term surgical success and complication rates were comparable to open surgery.

Annotated reviews are not available for download in order to protect the identity of reviewers who chose to remain anonymous.

---

## Round 0.2 · accepted · Accept

The authors performed appropriate revisions according to the reviewers comments.

Reviewer 1 ·

Basic reporting

Authors did changes accordingly the questions reviewers posed. The quality of manuscript was increased. I have no other questions.

Experimental design

-

Validity of the findings

-

Additional comments

-

Reviewer 2 ·

Basic reporting

I have no observations in this section.

Experimental design

I have no observations in this section.

Validity of the findings

I have no observations in this section.

Additional comments

It is where I consider it convenient to insert a paragraph and make corrections or minor modifications in the body of the manuscript, the text has been highlighted and note boxes have been placed in the PDF document (it is attached):

Material and methods:

Line 96 (USA).
Line 104 (add a paragraph): We did not include the postoperative stay to analyze it as a variable, because it was not valuable due to the hospital discharge policy of our institution.

Procedures:

Lines 117 and 125 (breakdown figure 1).

Results:

Add "also" on line 146.
On line 148 add space (14 laparoscopic).

References:

Correct in the 27 references, publication year order and add the DOI (digital object identification).

Annotated reviews are not available for download in order to protect the identity of reviewers who chose to remain anonymous.